# The State of the Art and Emerging Trends in the Wastewater Treatment in Developing Nations



Sangha Bijekar [1,2], Hemanshi D. Padariya [3], Virendra Kumar Yadav [2,4,*], Amel Gacem [5], Mohd Abul Hasan [6], Nasser S. Awwad [7], Krishna Kumar Yadav [8], Saiful Islam [6], Sungmin Park [9] and Byong-Hun Jeon [10,*]

1   Department of Biotechnology, School of Sciences, P P Savani University, Surat 394125, Gujarat, India
2   Research and Development Centre, YNC Envis Private Ltd., New Delhi 110059, India
3   JB and Karp Vidhya Sankul, Surat 394125, Gujarat, India
4   Department of Biosciences, School of Liberal Arts and Sciences, Mody University of Science and Technology, Lakshmangarh, Sikar 332311, Rajasthan, India
5   Department of Physics, Faculty of Sciences, University 20 Août 1955, Skikda 21000, Algeria
6   Civil Engineering Department, College of Engineering, King Khalid University, Abha 61421, Saudi Arabia
7   Department of Chemistry, King Khalid University, Abha 61413, Saudi Arabia
8   Faculty of Science and Technology, Madhyanchal Professional University, Ratibad, Bhopal 462044, Madhya Pradesh, India
9   Department of Civil and Environmental Engineering, Hanyang University, 222-Wangsimni-ro, Seongdong-gu, Seoul 04763, Korea
10   Department of Earth Resources and Environmental Engineering, Hanyang University, 222-Wangsimni-ro, Seongdong-gu, Seoul 04763, Korea
*   Correspondence: yadava94@gmail.com (V.K.Y.); bhjeon@hanyang.ac.kr (B.-H.J.)

**Abstract:** Water is the founding fundamental of life and hence is a basic need of life. However, due to the ever-rising population, industrialization has emerged as a global issue. This problem has notably escalated in developing countries. Their citizens face problems such as floods, drought, and poor water quality. Due to poor water quality and sanitation problems, most health issues are caused by water-borne infections. In developing countries, untreated wastewater is released into water bodies or the ground, thereby polluting natural resources. This is due to the lack of sufficient infrastructure, planning, funding, and technologies to overcome these problems. Additionally, the urbanization of megacities in developing countries is highly accelerated, but it is disproportionate to the required resources for treating wastewater. Due to this biological oxygen demand (BOD): chemical oxygen demand (COD) ratio is increasing exponentially in developing countries compared to developed ones. Spreading awareness, education and supporting relevant research, and making stringent rules for industries can alone solve the water problem in developing countries.

**Keywords:** point sources; unsustainable; urbanization; biodegradable; wastewater treatment

## 1. Introduction

No life form on earth can exist without water. Human civilization is dependent on water for domestic and industrial purposes. It is one of the most precious assets on this planet. The earth's surface consists of 71% water, but only 0.3% is usable (National Ground Water Association) as shown in Figure 1. Our water demand is fulfilled from underground, rivers, wells, and streams [1]. The earth is a closed system i.e., it does not allow the transfer or exchange of any matter including water. Hence, water is a limited resource. The water that was present billions of years ago is the same that we are using now. Mother earth maintains the water quantity and quality through the hydrological cycle. This substantiates the need of reusing water for our domestic and industrial purposes. Chemically, water is a universal solvent; it is tasteless, odorless, and colorless. Owing to this nature, water leaches minerals from nature that make it more beneficial for drinking purposes [2]. But due to the same characteristic, water also dissolves natural and artificial harmful chemicals

like cancer-causing metals, pesticides [3], detergents [4], and industrial wastes that may make it toxic and unusable. The quantity of such toxic matter present in water determines its quality.

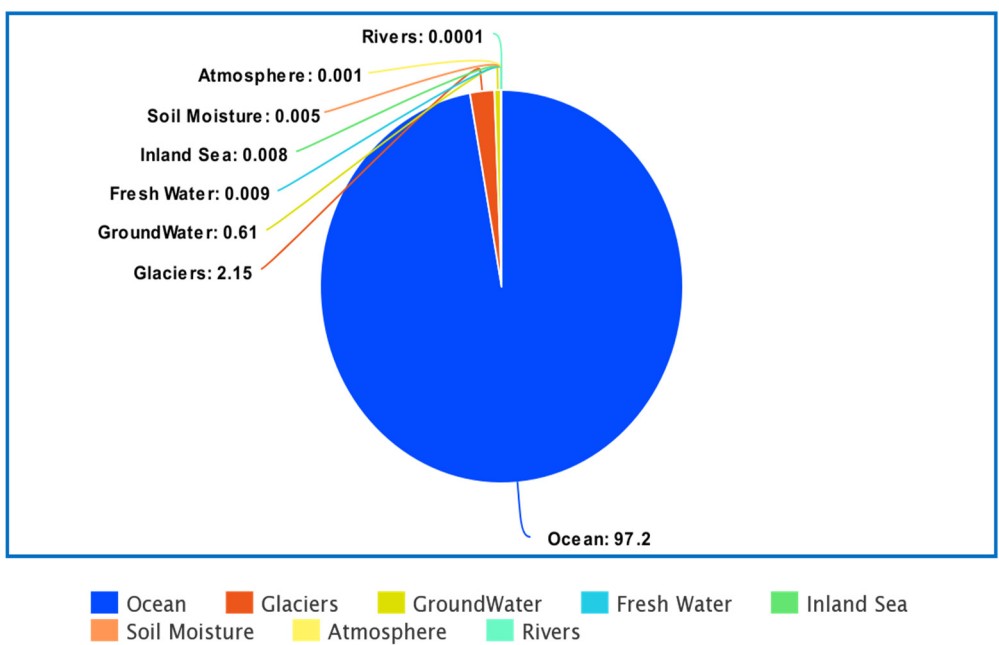

**Figure 1.** Distribution of earth water.

The organic and inorganic matter present in water makes it an ideal medium for microbial growth. The main cause of microbial contamination in natural water resources is the discharge of waste from domestics, organizations, hospitals, industries, etc. [5]. The use of untreated water could result in the declension of the ecosystem and public health. Contaminated water can transmit diseases such as diarrhea, cholera, dysentery, typhoid, and polio. It is estimated that drinking contaminated water causes 485,000 diarrheal deaths each year (WHO) [6,7].

Water pollution is increasing due to contemporary civilization accompanied by modern industries and the booming economic system. The direct and untreated discharge of used water from urban drainage and industries into rivers is a major cause of the degradation of water quality [8,9]. This is due to either lack of awareness, inappropriate drainage system, or unavailability of wastewater treatment plants [10–12]. Due to unawareness, residents dump their waste products into seas, streams, rivers, ponds, and lakes thinking of water bodies as the final home of waste material. Such human activities are not only responsible for spoiling water quality, but also for affecting aquatic life [13,14]. There is an urgent need to address these problems and employ the best water treatment methods. It is a challenge for humankind to make sustainable development in civilization without damaging the environment. When water is used for domestic and industrial purposes, the used water carries deleterious constituents. According to the United Nations (UN), about 80% of sewage is currently discharged without treatment [15–17]. The untreated water either sinks into the ground polluting groundwater, or flows into our natural resources and contaminates them [18–21]. Water management is a global issue [22,23]. Across the globe, countries are facing problems over the insecurity of water and water contamination. But the challenge of water quality and safety is more severe in developing nations. In order to reuse water efficiently, proper waste management policies and infrastructure are quintessential. Water scarcity and sanitation are global problems; however, they are more pronounced in developing nations [21]. In 1996, food and agriculture organizations (FAO) stated that the water quality programs that are run in countries collect the wrong sample from the wrong place using the wrong substrate, at wrong sampling frequencies, and produce unreliable

data and this is most commonly observed in developing countries (FAO, 1996). The water supply and sanitation, and the World Health Organization (WHO) have estimated that 25% of urban and 82% of the rural population of developing countries lack access to safe drinking water and sanitation services (CNES, 2003).

Ghosh et al., 2021 studied a comparative analysis of water systems in megacities in both developed, as well as developing nations. The authors concluded that improper water management is one of the major causes of morbidity and mortality due to disease [22].

Adelodun et al., 2021, assessed the socioeconomic inequality on the basis of consumption of water contaminated with viruses in developing nations. The authors revealed that there is high socioeconomic inequality in developing nations as compared to developed ones. The authors further revealed that the consumption of virus-loaded water usage may lead to waterborne diseases, which leads to several socioeconomic factors. In order to address the virus-contaminated water sources mitigation, strategies are proposed [24].

Chaitkin et al., 2022, studied the cost of wastewater treatment, sanitation, and hygiene in public health care facilities in about 46 of the least developed countries (as categorized by the United Nations). The investigators used a modeling study and concluded that a mean of $0.24–0.40 per capita, in capital investment is required every year, and annual operations and maintenance costs are expected to increase from $0.10 in 2021 to $0.39–0.60 in 2030 [25].

Before investigating this topic, authors searched water systems × developing nations, on science direct and found about 11,538 articles in 2022, 13,160 in 2021, 10,221 in 2020, 8547 in 2019, 8114 in 2018 and 7160 in 2017. Out of these five years (2017–2022), the maximum number were research articles i.e., 38,655, followed by 8272 book chapters, 7768 review articles, 1556 encyclopedia, and conference abstracts (95), and case reports were 65. So, from these statistics, the authors have concluded that in the last five years maximum research work in this area, which indicates that the water system in developing nations is one of the most important aspects in the whole world. So, in this review work, the authors have tried to find the latest happenings in this field and also find the existing research gap. Finally, the authors have suggested bridging the gap existing in this field.

The authors in the current review work have highlighted the current and emerging trends in wastewater treatment methods among developing countries. The authors have emphasized the recent advancement in wastewater treatment technologies in developed countries. They have also highlighted the status of various water resources in developed and developing nations. Finally, the authors have focused on the current and future challenges at the frontline of developing nations associated with water.

## 2. Sources of Wastewater

Wastewater is produced largely due to human activities. Water gets polluted from two major sources—point source (PS) and non-point source (NPS) as shown in Figure 2 [26]. When the source of pollutant is identifiable and it is released through the pipe, channels, or tunnels it is called a point source. Examples of PS are effluent released through pipes from homes, commercial areas, hospitals, wastewater treatment, and industries. Whereas, when pollutant enters the water resources via unidentifiable sources then it is known as NPS [26,27]. Examples of NPS are run-offs from agriculture, forests, mines, stormwater, etc.

Monitoring and regulating the non-point source of water pollution is one of the biggest challenges for the scientific community and authorities [28,29]. Dealing with the non-point source of water pollution is analogous in developing and developed countries [30]. However, it is more conspicuous in developing countries than developed countries due to the insufficient infrastructure, unplanned and unsustainable urbanization, and lack of updated technology [31–33]. Table 1 is showing various types of sources of water pollution.

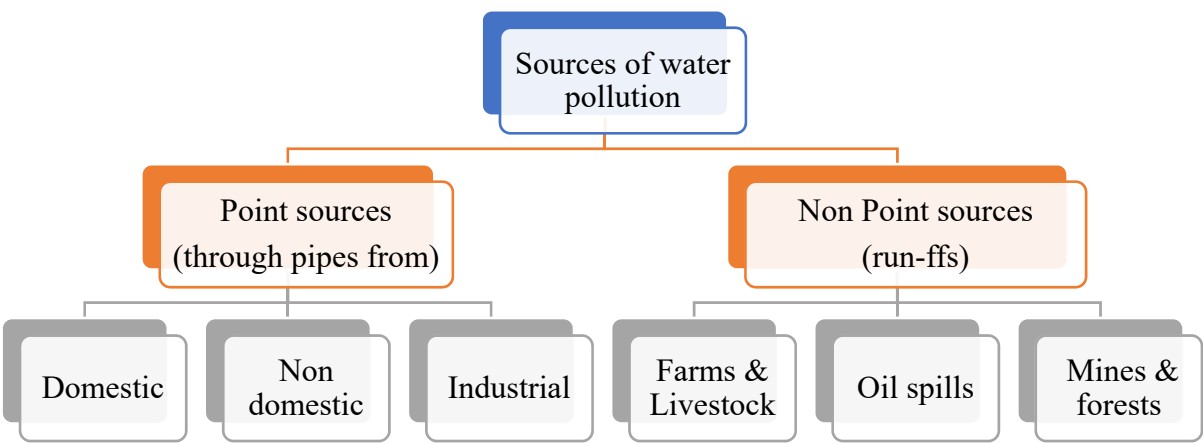

**Figure 2.** Sources of Water Pollution.

**Table 1.** Sources of water pollution along with their side effects.

| Sources | Pollutants along with Their Effect | References |
|---|---|---|
| Waste generated from agricultural run-offs | Phosphates and nitrates in fertilizers Causes eutrophication | [34] |
| | Pesticides, herbicides | [35,36] |
| Animal waste from farms | Microorganisms (bacteria, protozoa, and viruses) that may cause water-borne diseases | [37] |
| Untreated sewage (human fecal matter + domestic waste) | Suspended solids and microorganisms Suspended solids may lead to reduced sunlight penetration in lakes, ponds other water bodies affecting aquatic flora and fauna. Pathogenic microbes from carriers may cause food and water-borne diseases like cholera, typhoid, etc. | [38,39] |
| Industrial effluents (metal-based, dyes, rubber, etc.) | Heavy metals like As, Hg, Cu, Cd, Cr, and Ni may accumulate in aquatic organisms and enter the food chain, leading to bioaccumulation. Severe to chronic diseases caused by heavy metals. | [7,8,40–42] |
| Underground pipes | Pb Highly toxic heavy metals may cause mental disorders among children | [43,44] |
| Domestic waste | Detergents, surfactants, may cause foaming of water bodies leading to reduced water supply to aquatic organisms. | [4,45] |

## 3. Composition of Wastewater

The world's freshwater resources are becoming increasingly polluted with organic waste, pathogens, fertilizers and pesticides, heavy metals, and emerging pollutants [46]. The organic and inorganic matter released from point and non-point water polluting sources is causing a reduction in river dilution capacity [47]. Domestic wastewater is discharged from homes and communities. Domestically, water is used for washing, cleaning, and flushing. Hence, used water carries soluble and non-soluble solid material. The domestic wastewater released due to washing, laundering, bathing, or showering is called Grey Water. The Domestic wastewater released from toilets or urinals is called Black Water [48] (Environmental protection Agency, 2021). The black wastewater is rich in organic matter and mostly biodegradable, it acts like nutritional media for pathogenic microbial growth, and hence it becomes hazardous and septic [49–51].

Industrial wastewater is an aqueous discharge released by industries. Industries need water for manufacturing, processing, cooling, diluting, washing, or transporting the product. The used water from industries carries suspended or dissolved matter released during industrial processing [52–54]. The content and quality of industrial wastewater

depend on the type of industry and its product [55]. The quantity of wastewater generated in many industries varies substantially from process to process and is substantially higher in developing countries [56].

The major pollution contributors are manufacturing industries like power generation, mining and construction, and food processing industries [8,45,57,58]. It may contain biodegradable [3,59], non-biodegradable [60,61], synthetic or heavy metal matter [62]. Developing countries are facing an extremely miserable situation where more than 90% of sewage waste and almost 70% of industrial raw wastes are becoming a part of the water bodies [63]. In developing countries, industrial waste disposal becomes more challenging because of the high production of wastes per unit area and the decrease in the proportion of land available for its disposal. Most industries dispose of raw effluent, which contains pathogens, heavy metals, soluble and insoluble salts, and organic and inorganic matter causing groundwater pollution [64–66]. Very few industries have adequate wastewater treatment facilities. As a result of this, freshwater becomes unfit for human use.

The wastewater composed of sewage; industrial, and agricultural waste is discharged globally in tens of millions of cubic meters per day. As a result, freshwater resources are polluted with fertilizers, pesticides, organic wastes, heavy metals, pathogens, agriculture waste, and emerging pollutants [57,67]. The acute poisoning of pesticides is causing morbidity and mortality, chiefly in developing countries (WWAP, 2018). The emerging pollutants are those, which have been recently identified as a danger to human health (UN, 2020). The emerging pollutants include chemicals and drugs used in modern lifestyles like cosmetic and personal care products, pharmaceuticals, hormones, endocrine disrupting chemicals, and pesticides [68–70].

According to reports, in developing countries untreated wastewater is discharged into rivers and water bodies, resulting in endangerment of aquatic life. The natural or synthetic pollutants, which are not commonly monitored in the environment, have the potential to invade the environment and cause a detrimental effect on human health and ecology. The European Aquatic Environment (EAE) has listed more than 700 emerging pollutants. Normal Network has categorized the emerging pollutants into more than 20 classes based on their origin. The major classes are pharmaceuticals, pesticides, disinfectants, and chemicals from various industries [71].

In developing countries, there were no studies regarding emerging pollutants due to a lack of awareness, funding, and infrastructure. As a result, the health risk associated with wastewater use is higher in developing countries. After 2005 and subsequently, research in this area was initiated, however, much more research and studies need to be conducted in developing countries [72,73]. Table 2 shows various types of pollutants and their composition, released from industries and domestic sources.

**Table 2.** Industrial and domestic and their effluent composition [74].

| Major Sources | Source | Composition of Effluent | References |
|---|---|---|---|
| Industries | Petroleum, stainless steel, paint industries, power generation, mining, construction, and food industries. | Zinc, copper, nickel, cadmium, xylene, benzene, toluene, NORM, monocyclic aromatic hydrocarbons, chromium, lead, mercury, aluminum, radioactive molecules, platinum group metals, silver, barium, silver, arsenic, molybdenum, chloramines, radium, fluoride, nonylphenol ethoxylates (NPE), di-(2-ethylhexyl)phthalate, polycyclic aromatic hydrocarbons, polychlorinated biphenyls, polychlorinated dibenzo-p-dioxins, and polychlorinated dibenzo-p-furans, nitro musks (chloronitrobenzenes), oestrogenic compounds and polyelectrolytes, etc. | [75–78] |
| Domestic | Kitchen and sewage. | Fertilizer and pesticides, lead and cadmium (from batteries), plastic bags, papers, polythene bags, rotten fruits and vegetables, and linear alkyl benzene sulphonates (LAS) are the major components of washing powder and found in detergents and microbes, etc. | [79–81] |

## 4. Hazardous Effects of Pollutant Consumption on Human Health

The increase in contaminants and pollutants in drinking water and their potential to alter human health is a concerning issue across the globe [82]. Compared to industries, domestic wastewater is relatively less hazardous [83]. As per WHO and the World Water Development Report (WWDR), the use of pesticides has caused significant morbidity and mortality, and it is more perturbing in developing countries [84]. Moreover, pathogen-related mediation, a new health risk assessment, and mitigation steps are required in developing countries [85]. Table 3 compiles the hazardous health effect caused due to consumption of pollutants via water.

**Table 3.** Hazardous health effects caused due to consumption of pollutants via water.

| Types of Pollutants | Health Hazard | References |
| --- | --- | --- |
| **Inorganic pollutants** | | |
| Zinc | Causes nausea, vomiting, and stomach cramps. | [86] |
| Copper | Vomiting, diarrhea, stomach cramps, nausea, liver damage, and kidney disease. | [87] |
| Nickel | Alters the gastrointestinal tract, respiratory system, skin, heart, and lungs; may cause epigenetic effects. | [88] |
| Cadmium | Affects the lungs, kidneys, and bone, causes fever, chills, and body pain. | [89] |
| Chromium | Mutagenic and carcinogenic. | [90] |
| Lead | Causes constipation, loss of appetite, headache, abdominal pain, causes tiredness and weakness; may lead to memory loss. | [91] |
| Mercury | Damages the kidneys, brain, vision, hearing capacity, and developing fetus. | [92] |
| Aluminum | Alzheimer's disease. | [93] |
| Platinum group metals | Carcinogenic. | [94] |
| Silver | Skin discoloration. | [95] |
| Barium | Slows the respiratory, digestive, and cardiovascular system. Damages kidney, liver, and spleen. | [96] |
| Selenium | Causes dizziness and weakness. May lead to prostate cancer. | [97,98] |
| Uranium | Induces cancer and is toxic to the kidneys. | [99] |
| Radium | Carcinogenic. | [100] |
| Fluoride | Causes bone disorders. | [101] |
| **Organic pollutants** | | |
| Xylene | Influences the central nervous system (CNS) and vision. | [102] |
| Benzene | Induces vomiting, dizziness, sleepiness, and heart problems; carcinogenic. | [103] |
| Toluene | Impairs CNS causing fatigue and drowsiness. | [104] |
| NORM | Carcinogenic. | [105] |
| Chloramines | Can cause hemolytic anemia. | [106] |
| Linear alkylbenzene sulphonates (LAS) found in detergents | May cause colon cancer. | [107] |
| Polycyclic aromatic hydrocarbons | Consumption leads to cancer. | [108] |
| Polychlorinated biphenyls | Carcinogenic in nature. | [109] |
| Polychlorinated dibenzo-p-dioxins and Polychlorinated dibenzo-p-furans | May cause malfunctioning of the immune and endocrine systems affecting reproductive health. | [110] |
| Nitro musks (chloronitrobenzenes) | Disturbs endocrine and gynecological health. | [111] |
| Oestrogenic compounds | Affects the normal functioning of the endocrine system. | [112,113] |
| Polyelectrolytes | Affects the gastrointestinal tract and lymphoid system. | [114] |
| Nitrate | Colorectal cancer, thyroid disease, and neural tube defects. | [115] |
| Pesticides | Affects the brain and induces Parkinson's disease, affecting reproductive and respiratory health. May be carcinogenic. | [116] |

**Table 3.** *Cont.*

| Types of Pollutants | Health Hazard | References |
|---|---|---|
| Pathogens | Health Hazard. | Reference |
| *Campylobacter jejuni* | Gastroenteritis. | |
| *Escherichia coli* | Gastroenteritis. | |
| *Salmonella* spp. | Salmonellosis, typhoid, parathyroid. | |
| *Shigella* spp. | Bacillary dysentery. | |
| *Vibrio cholerae* | Cholera. | |
| *Yersinia* spp. | Gastroenteritis. | |
| Adenovirus | Upper respiratory infection and gastroenteritis. | |
| Astrovirus | Gastroenteritis. | |
| Coxsackie A virus | Meningitis, pneumonia, fever. | |
| Echovirus | Meningitis, paralysis, encephalitis, fever. | |
| Hepatitis A virus | Infectious hepatitis. | |
| Hepatitis E virus | Infectious hepatitis, miscarriage, and death. | |
| Human calicivirus | Epidemic gastroenteritis with severe diarrhea. | |
| Polio virus | Poliomyelitis. | [117] |
| Reovirus | Respiratory infections, gastroenteritis. | |
| Rotavirus | Acute gastroenteritis with severe diarrhea. | |
| TT hepatitis | Hepatitis. | |
| *Balantidium coli* | Balantidiasis. | |
| *Cryptosporidium* spp. | Cryptosporidiosis. | |
| *Entamoeba histolytica* | Acute amoebic dysentery. | |
| *Giardia duodenalis* | Giardiasis. | |
| *Toxoplasma gondii* | Toxoplasmosis. | |
| *Ascaris lumbricoides* | Ascariasis. | |
| *Ascaris suum* | Coughing and chest pain. | |
| *Hymenolepis nana* | Hymenolepiasis. | |
| *Necator americanus* | Hookworm disease. | |
| *Taenia saginata* | Insomnia, anorexia. | |
| *Taenia solium* | Insomnia, anorexia. | |
| *Toxocara canis* | Fever, abdominal pain, muscle ache. | |
| *Trichuris trichiura* | Diarrhea, anemia, weight loss. | |

In 1992, the WHO also reported that in developing nations, 3.2 million children under the age of five die every year due to impecunious sanitation and unsafe drinking water. The WHO (2019) has estimated that around 829,000 people would die each year due to diarrhea caused by unsafe water. According to the World Bank (1993), infections caused by drinking contaminated water are the third leading reason for morbidity and mortality in developing countries [118,119]. Pathogen contamination is the most widespread water quality problem in developing countries due to unsafe water and sanitation [120,121]. Emerging pollutants present a new global water quality challenge in both developed and developing countries, with potentially serious threats to human health and ecosystems.

Gleick (2002) predicted that if proper measures were not taken about wastewater, 2,076,752 people would die by 2022 due to water-related diseases. He had forewarned about the issue and had affirmed the need for attention to public health. To counterbalance the adverse effects of untreated wastewater, developing countries should address the issues by investing in wastewater treatment technology [122].

The influx of wastewater into water bodies not only affects human health but also causes a negative impact on the environment and ecology [123]. The accumulation of organic and inorganic pollutants and their microbial degradation decreases dissolved oxygen, altering aquatic life and reducing the survival rate of fishes and other life forms [124]. The nutritional load of wastewater causes eutrophication, which further causes invertebrate biodiversity loss, and decreases cross-taxon congruence; this was confirmed by Wang in 2021 [125]. This also reduces the penetration of sunlight, minimizing photosynthesis and inhibiting plant growth. The wastewater also alters the physical characteristics of water, like temperature, affecting potential ecological impacts [126].

## 5. Population Growth and Scarcity of Safe Drinking Water in Developing Nations

Access to clean drinking water is a human right. The supply of quality water and standard of sanitation determines the quality of life. But in developing nations like India, Nigeria, Pakistan, Bangladesh, and most of the African countries, etc., people lack this basic accessibility. An increase in population increases water demands, thereby producing more wastewater. Roland Berger Strategy Consultants conducted research and estimated that developing regions will see 97% of the world's population with a growth of 1.2 billion people between 2013 and 2030, Liyanage and Yamada have identified a direct relationship between population and wastewater production [127]. While the world's population growth rate has slowed since the 1980s, population numbers are still growing fast, in particular in developing countries (FAO water reports 2012). With the increase in population, the management of urban wastewater is a critical issue for urban planners. There is also a need to think of domestic wastewater as a resource, which can be utilized after treatment for meeting the non-potable requirements. It will help in conserving raw water resources like groundwater and surface water according to the Central pollution control board (CPCB) [128]. In developed nations, the government is committed to covering almost 99% of sanitation and considers it a part of public services, whereas, in developing nations, it covers only 50% [129,130]. To a great extent, recently, parts of the world have already started feeling the "water crunch". It is believed that by 2025, India, China, select countries of Europe, and Africa will face water scarcity, whereas Figure 3 is showing Water withdrawal as a percentage of the total available.

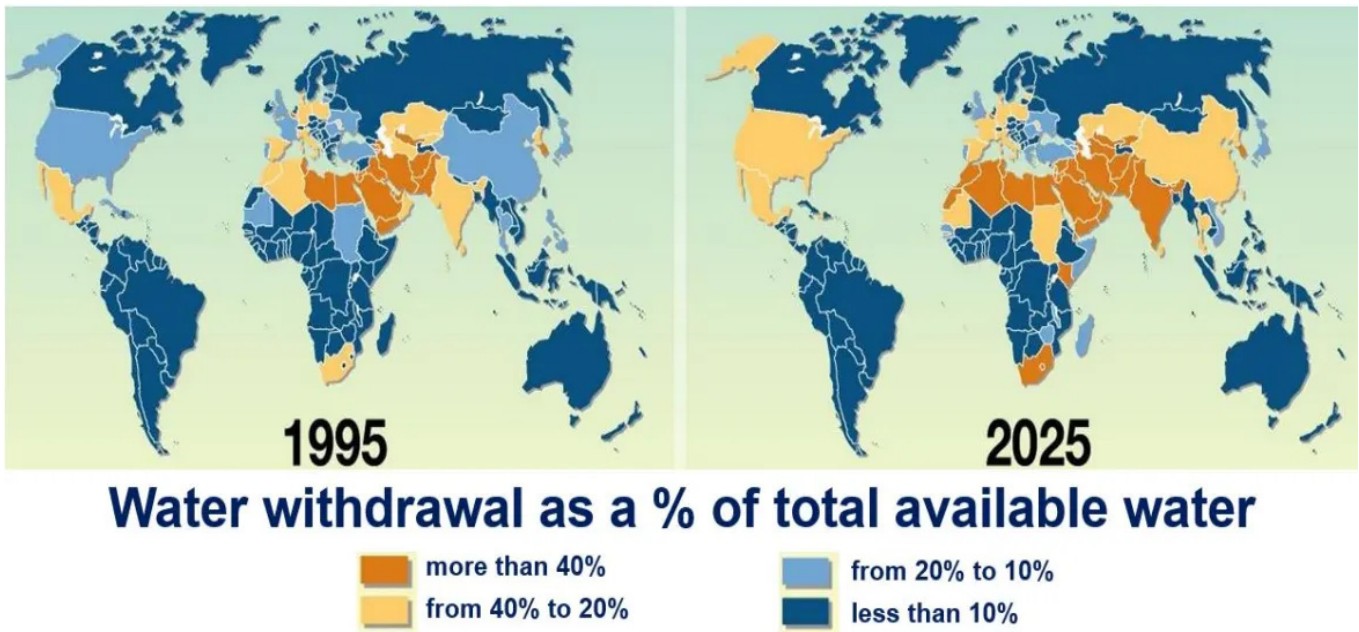

**Figure 3.** Water withdrawal as a percentage of total available.

In developing nations, water pollution and safe drinking water are major challenges for citizens. Urbanization is a significant pollution source, particularly in developing countries, especially for groundwater, as a result of under-managed solid waste disposal and poorly managed sanitation infrastructure (UNESCO, UN-Water, 2020). About 80 to 90% of developing nations neither collect the wastewater nor treat it [131] (UNESCO, UN-Water, 2017), the emissions related to water supply and the sanitation sector—and its potential to contribute significantly to climate change mitigation—should not be neglected.

Due to the huge population, the challenge becomes more critical. Due to the low economy, lack of awareness, incompetent policies, and absence of a political will, the water problem is still baffling. As a result, it is also affecting citizens' health and the environment of developing nations. The scarcity of water is a critical issue and thus the developing nations must set a goal to work to strengthen policy, bring awareness, and invest in the wastewater management system (UNESCO). The developing nations need to work on the supply and reuse of water in order to improve the productivity and quality of life of their citizens.

## 6. Urban Water Cycle in Developing Nations

To understand the recycling and reuse of water, it is important to know how the water moves in the earth system (NRC, 2012). The Earth's atmosphere, lithosphere, biosphere, and anthroposphere communicate via the water cycle. Their interaction is strongly affected by human activities and socio-economic development [132]. Earth stores water in rivers, oceans, soil, streams, glaciers, snowfields, environment, and groundwater as shown in Figure 4. Water is recycled by transpiration, condensation, precipitation, percolation, snowmelt, and runoff. Aquifers get recharged and refilled by precipitation.

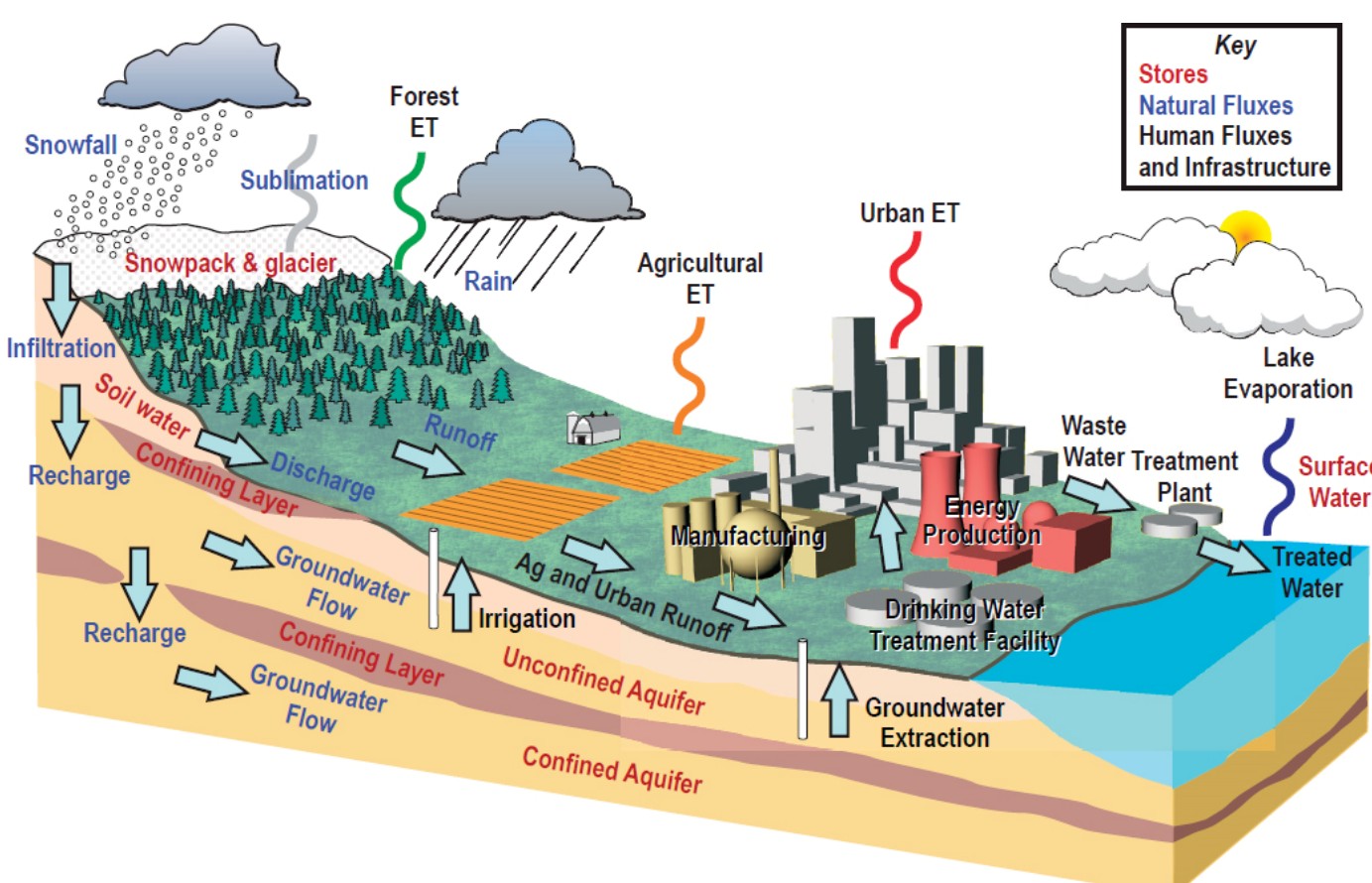

**Figure 4.** Global water cycle, source: Courtesy of Jerald Schnoor.

Urbanization is a part of the economic development of most countries, so it is not surprising that developing countries, where per-capita income is less, also had faster population growth in cities [133]. Population, urbanization, and industrialization affect the natural water cycle. In urban areas, the construction of water supply and drainage discharge disturbs the natural process. In urban areas, water moves from rivers, lakes, and wetlands to homes and industries. Using an electric current, water is made to flow upward, which is against the natural law [134]. Thus, because of anthropogenic activities, the water cycle becomes more complex in urban areas. Hence, the modified cycle is called the urban water cycle [135]. The urban water cycle consists of two major sources of water: municipal water supplies and precipitation [136]. For sustainable development, coordinated development of the urban water cycle is the key [137] (Hoekstra et al., 2018).

In developing countries, the process of urbanization is rapid, and it accounts for disproportionately millions of residents of megacities. Urbanization demands high quantities of raw material, energy, and water, and in return, the developing cities produce a large number of pollutants affecting the water resources [138]. In developing countries, the cost involved in water supply services and technological development affect the level of water demand. During the next 50 years, the population will increase significantly in the megacities of the world. Many of these megacities are from developing countries (UN-Habitat, 2006). Figure 5 shows the water cycle in an urban region.

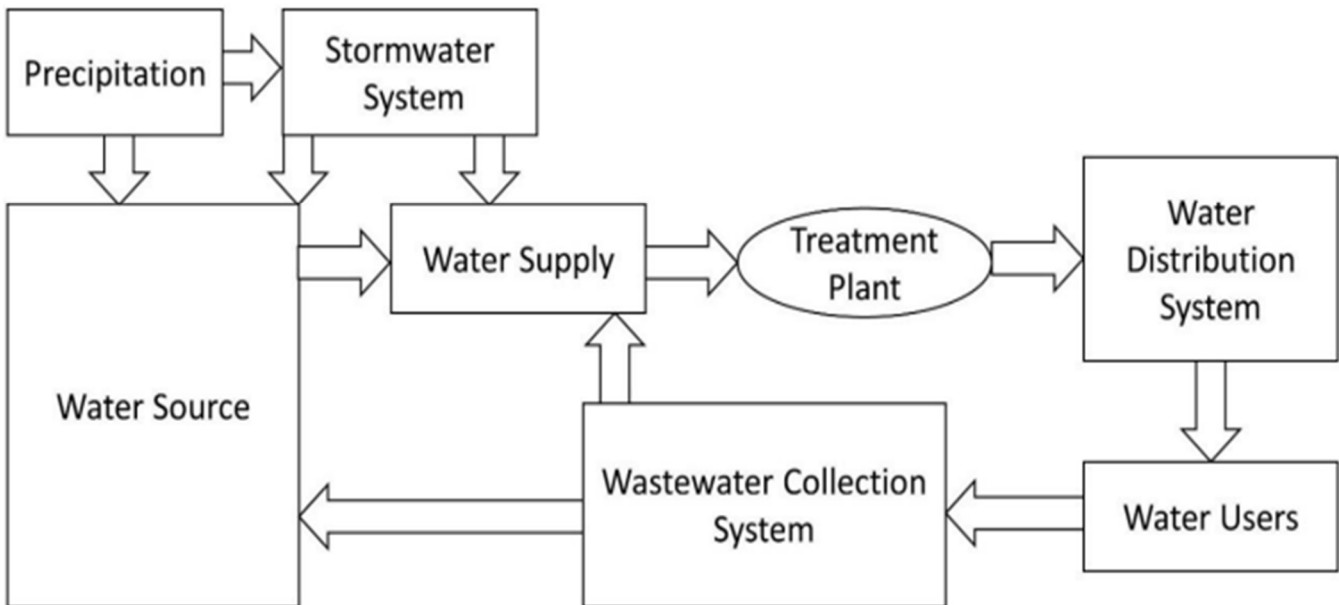

**Figure 5.** Water cycle in the urban region [138].

The urban water cycle consists of two water resources—municipal corporation water supply and precipitation. The first source i.e., municipal water, which is imported from natural resources or dams, is far from the urban areas. During the transport of water, some of the water is lost or leaked into urban groundwater. The received water is used by the urban populace, which then produces wastewater. The wastewater discharges and goes to surface waters. The second source is precipitation. Due to urbanization, the received rainwater may get subjected to hydrological abstractions or may run off (non-point source) and reach natural or man-made water storage systems [138].

## 7. Funding for Wastewater Treatment in Developing Countries

The gap between the developing and developed world is increasing with the passage of time. There is also a decline in providing aid, which goes to funding wastewater treatment too. Considering the population explosion and water scarcity in developing countries, the Food and Agriculture Organization of the United Nations has anticipated that by 2050,

93 developing countries would need US$960 billion for developing water infrastructure and improving irrigation [139], (UNESCO, UN-Water, 2020). Considering this scenario, both developing and developed countries need to make an investment in wastewater treatment plants. But as per the UNESCO, UN-Water, (2017) report, due to the presence of wastewater, the BOD: COD ratio is undoubtedly lower in developed countries than in developing countries. Hence, it is more compelling to invest in developing countries due to the lack of data related to wastewater management and water quality [140]. According to the UN, data plays a pivotal role in bringing positive change in the social and economic environment of the private and public sectors of the country. It also supports making decisions and promotes political will, and investment [141]. The extensive problem of wastewater in developing countries is due to the lack of data related to water quality.

Wastewater treatment plants need funding for the construction of required infrastructure, technology, and maintenance. Developing countries do not have the local expertise and sufficient funds to operate the plants efficiently [142]. Wastewater treatment needs high investment infrastructure, which is not affordable in developing countries (UNESCO, UN-Water, 2017).

According to Das Gupta, low economy, poor administration, and a dearth of knowledge about water demand and availability hinders development. The total investment of GDP for electricity, water, roads, and communication in 1996, in developing countries, was US$230 billion (Sunman1999 and Briscoe 1998). Out of the total investment, only $25.3 billion was invested. To promote wastewater management programs in such countries, their economic feasibility must be assessed [143].

As per Serageldin (1994), developing countries have two agendas to protect against environmental health challenges. The first one is to provide adequate sanitation to houses, and the second is to manage and reuse urban wastewater and protect the natural water resources from getting polluted. Along with this, the industries in such countries also need to work with the consent of environmental health and safety policies. The currently used primary and advanced techniques in treating wastewater are unsatisfactory and hence it is an urgent need to promote and upgrade the technology. However, for developing countries, low-cost, effective, and appropriate technologies are needed in order to reuse wastewater [144].

Along with infrastructure, developing countries must also invest in spreading awareness, educating people, building collaborations, and improving governance for sustainable development. According to the World Bank, funding issues in wastewater treatment can be resolved by a public-private partnership. They confer that the involvement of the private sector in water sanitation would strengthen with time. But they are not aware of its impact in such countries. Considering the conditions of developing nations, the conventional method for wastewater treatment doesn't seem to be appropriate. Hence an alternative approach is needed.

## 8. Centralized and Decentralized Wastewater Treatment in Developing Countries

There are two approaches for wastewater treatment—centralized and decentralized. The centralized approach is a conventional one, where wastewater is treated at a central location and then distributed using a network of the pipeline network. The centralized system uses a combination of different processes and techniques like flocculation, coagulation, filtration, sedimentation, disinfection, etc. The techniques and processes ensure maximum removal of organic matter, pollutants, and pathogenic microbes. The major disadvantage of a centralized system is the need for huge investment in building the infrastructure and its maintenance. Decentralized wastewater treatment uses a range of simple technologies to treat wastewater from near or at the point of generation, for example, individual homes, colonies, institutes, or industries. Considering the problem of non-point source water pollution, decentralized wastewater treatment seems to be a logical and sustainable option. According to [145], a centralized wastewater treatment plant is more expensive than a

decentralized one. The major differences between centralized and decentralized are shown in Table 4.

**Table 4.** Centralized vs. decentralized Source—[146].

|  | Centralized | Decentralized |
|---|---|---|
| Definition. | The wastewater is collected and treated at a central location and then distributed using a network of a pipeline network. | Decentralized wastewater treatment is the use of a range of simple technologies to treat wastewater from near or at the point of generation. |
| Also called: | Off-site Treatment. | On-site Treatment. |
| Huge network of pipeline, excavation, manholes. | Required. | Not required. |
| Length and diameter of pipes. | Large and long. | Small and Short. |
| Waste is collected from | From long distance. | Near or at the site of a pollution source. |
| Composition of collected wastewater | A mixture of black, grey, and industrial water. | The type of water is separated from the source. |
| Owned by | State Authority | |
| Scale of operation | Large. | Small. |
| Area required | A huge area is required in a central place. | A small area is required at the site of the pollution source. |
| Investment | Huge investment is required. | Comparatively less and it is cost-effective. |
| Technical Staff | Dedicated staff are required for maintenance | Can be maintained fortuitously. |
| Preferred for | High population density region | Low population density region. |
| Affordability | People of developed countries can afford. | People in developing countries cannot afford. |
| Technology used | Combination of different advanced types of technologies | Primitive technology like septic tank/drain field. |
| A consequence of improper maintenance | Detrimental health issues at the regional level | Detrimental health issues in local areas. |

Laugesen et al. (2010), have predicted that "despite the past failures of most centralized systems, it is likely that most new wastewater management systems in developing countries will continue to be advanced, centralized, and with a continued high probability for failure" [147]. UNESCO, and UN-Water (2017), have alarmed such countries about the risky investment in the centralized system due to inadequate finance and institutional capacity. As per USEPA research, decentralized wastewater management is more appropriate for developing countries due to its cost-effectiveness [145]. The lack of awareness, education, and research in such countries failed to select the appropriate approach for wastewater treatment. They also fail to consider the geographical conditions, human resources, cultural aspects, and finance.

In 2018, Zinn and his coworkers reviewed different technologies used for wastewater treatment in developing countries. They enlisted technologies, namely solar water disinfection (SODIS), chlorination, ceramic and bio-sand water filtration, and slow sand filtration. They compared the strength and weaknesses of these technologies, and also mentioned the pathogenic bacteria that the respective technology eliminates. They also found that SODIS is a more efficient technology than others, but hypothesized that in the future, membrane filtration will be preferred. The authors have also compared the effectiveness, advantages, and disadvantages of these technologies.

In India, the new wastewater treatment technology is a combination of electrocoagulation and electroflotation [148]. Electrocoagulation uses an electrical charge to change the surface charge of the particles due to which the particles become aggregated. In electro flotation, the suspended particles are removed by passing electricity through water generating hydrogen and oxygen [149]. In African countries, technologies like sludge deactivation, trickling filter, oxidation ditch, and aerated lagoons treat the wastewater [150,151]. China has come up with electron beam technology to treat industrial and medical wastewater. China has started the World's largest wastewater treatment facility [152]. In Turkey,

technologies like UV disinfection, sand filter, chlorination, rapid sand filter, disc filter, membrane bioreactor, activate carbon, ultrafiltration, ozone disinfection, and cartridge filters are used. Turkey is facing technical, social, and economical barriers to reusing wastewater [153].

## 9. Challenges

From the various pieces of literature, it has been found that there are several challenges faced by developing nations in terms of wastewater and sanitation. Scarcity of water is one of the major challenges among all such nations. Besides this, the unusual flood and its consequences lead to the siltation problem and river and dam contamination which gives rise to source receptor issues. Besides this, there are problems associated with poor access to water and poor water resource management. The quality of water could be greatly affected by the water productivity in the agricultural sectors. Water affordability is another issue, in addition to its storage, awareness, and prevention of cross-contamination of potable water with pollutants. An integrated approach is required in developing countries along with proper management of solid waste and wastewater to enhance the quality of potable water. Water conservation is another issue that could be solved by awareness.

## 10. Conclusions

Poor water quality and water shortage is a global problem in this era but it is notably severe in developing countries. Third-world countries need to urgently upgrade their technologies for treating wastewater and promote alternative approaches. They also need to redefine their policies and make more stringent rules and regulations for industries discharging waste. Government must promote research on the recycling and reuse of wastewater. The developing countries should invest in the decentralized system of water treatment, and they should actively participate in 17 sustainable development goals proposed and missioned by the United Nations. The developed countries should lend a helping hand and support such countries in managing their water resources and for their sustainable use. The Government and policymakers must prioritize the development of the civil and IT infrastructure required for wastewater treatment. And lastly spreading awareness and educating people could be the best solution.

**Author Contributions:** Conceptualization, S.B., H.D.P., V.K.Y. and N.S.A.; Data curation, V.K.Y., S.I., K.K.Y., A.G. and S.B.; methodology, S.B., H.D.P., M.A.H., S.P. and B.-H.J.; validation, K.K.Y., S.P., B.-H.J. and V.K.Y.; formal analysis, S.B., N.S.A., V.K.Y., H.D.P. and A.G.; resources, K.K.Y., B.-H.J., M.A.H. and S.I.; writing—original draft preparation, S.B., V.K.Y., K.K.Y., B.-H.J., H.D.P. and A.G., writing—review and editing, S.B., H.D.P., V.K.Y., S.P., A.G., N.S.A., M.A.H. and S.I.; supervision, V.K.Y., S.B., A.G., S.I. and B.-H.J.; project administration, V.K.Y., K.K.Y. and N.S.A.; Funding acquisition, B.-H.J., S.I., N.S.A., S.P. and M.A.H.; Investigation, S.B., V.K.Y. and B.-H.J.; Software's, K.K.Y., M.A.H. and N.S.A.; Visualization, S.P., V.K.Y. and M.A.H. All authors have read and agreed to the published version of the manuscript.

**Funding:** This research received no external funding.

**Institutional Review Board Statement:** Not applicable.

**Informed Consent Statement:** Not applicable.

**Data Availability Statement:** All relevant data are included within the article.

**Acknowledgments:** The authors extend their appreciation to the Deanship of Scientific Research at King Khalid University, Abha, Kingdom of Saudi Arabia for funding this work through Large Groups RGP.2/43/43. This work was supported by the Korea Environment Industry and Technology Institute (KEITI) through Subsurface Environment Management (SEM) Projects, funded by Korea Ministry of Environment (MOE) (No. 2020002480007).

**Conflicts of Interest:** The authors declare no conflict of interest.

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
