# Peer review of "The State of the Art and Emerging Trends in the Wastewater Treatment in Developing Nations"

_water, doi:10.3390/w14162537_

Round 1

Reviewer 1 Report

In this study, the authors reviewed the water resource, sources and present situation of wastewater, and control strategy of wastewater in developing countries. The manuscript was clearly organized and the summary was complete, the conclusions were reliable, and some good suggestions are given to the existing problems. However, some revisions were needed prior to a possible publication in this journal.

1. In the 5th section, it is suggested to add some brief descriptions of ecological impacts.

2. Table 2, the first column is about “Inorganic Elements”, however, the contents include inorganic elements, organic matter and pathogenic bacteria; besides, many references cited are older.

3. Line 189, what are the typical new pollutants?

4. Line 223, “80%”.

5. Line 312, don't use the first person.

6. In 9th section, the methods and practical technologies for treating wastewater in developing countries need to be described in detail, as well as the advantages and disadvantages of the technology; besides, give some typical application cases in developing countries.

7. What are the trends in wastewater treatment for developing countries? Please describe in detail.

Author Response

In this study, the authors reviewed the water resource, sources and present situation of wastewater, and control strategy of wastewater in developing countries. The manuscript was clearly organized and the summary was complete, the conclusions were reliable, and some good suggestions are given to the existing problems. However, some revisions were needed prior to a possible publication in this journal.

  1. In the 5th section, it is suggested to add some brief descriptions of ecological impacts.

A/R: Thank you for these valuable comments and suggestions. The authors have now added brief descriptions of ecological impacts in the 5th section as suggested by the reviewer in the revised version of the manuscript.

  1. Table 2, the first column is about “Inorganic Elements”, however, the contents include inorganic elements, organic matter and pathogenic bacteria; besides, many references cited are older.

A/R: Thank you for this valuable comment and suggestions. The authors have now separated inorganic, organic elements, and pathogenic microbes into separate columns as suggested by the reviewer. The author said also I did recent references in the table as well as in the manuscript as suggested by the reviewer.

  1. Line 189, what are the typical new pollutants?

A/R: Thank you for this valuable comment and suggestions. The authors have added all the typical new pollutants in the revised manuscript as suggested by the reviewer.

  1. Line 223, “80%”.

A/R: Thank you for this valuable comment and suggestions. The authors have now rectified the mistake in the revised manuscript as suggested by the reviewer.

  1. Line 312, don't use the first person.

A/R: Thank you for this valuable comment and suggestions. The authors have now rectified all such mistakes from the revised version of the manuscript as suggested by the reviewer.

  1. In 9th section, the methods and practical technologies for treating wastewater in developing countries need to be described in detail, as well as the advantages and disadvantages of the technology; besides, give some typical application cases in developing countries.

A/R: Thank you for this valuable comment and suggestions. The authors have now described the said section in the revised manuscript as suggested by the reviewer.

  1. What are the trends in wastewater treatment for developing countries? Please describe in detail.

A/R: Thank you for this valuable comment and suggestions. The authors have now added trends in wastewater treatment for developing countries in the revised manuscript as per the suggestions of the reviewer.

Reviewer 2 Report

In this manuscript, the authors compiled interesting information from the literature about wastewater. They described its sources, composition, impact on human health and treatment. The review was divided into 10 sections, only one of which (section 9) was focused, but not extensively, on wastewater treatment. In another section (8) the authors discussed the funding aspect. Based on the title, the manuscript should mainly and exhaustively review recent trends in wastewater treatment, which, unfortunately, was not the case. The text does not read well and contains many grammar errors. There is no smooth connection between sentences in many paragraphs (same as reading different notes from the literature). I would recommend an extensive revision of the text prior to publication.

Abbreviations and acronyms used in the text were not defined. The authors should either define each abbreviation/acronym prior to first use in the text or provide a list of the abbreviations and their corresponding definitions.

Grammar revision is needed all along the text (e.g., verify text in lines 108-109, 110, 129-130, 152, 158, 161, 167-171, 198, 203, 232, 296, 311, 312, 326, 336, etc.)

In some cases, “developing countries” was too much repeated in a same paragraph. For examples in lines 355-356, it reads in a row: “The developing countries should invest… The developing countries should actively participate …”. Some rewording and good grammar could help avoiding such unnecessary repetition.

Since the review is about the "state of the art" of wastewater treatment, the section 2 is redundant.

There are two figures 3 (Figure 3. Sources of Water Pollution, and Figure 3. Source – United Nation Population Fund.)

Lines 108-109: incomplete sentence

Line 110: "commercials" means advertisements. Also, instead of "... hospitals wastewater treatment and Industries.", I think you mean "... hospitals, wastewater treatment plants, and industries."

Line 111: "... run off from like run off from agriculture, ... ". Consider revision.

Table 1: title is about the composition of industrial effluent, but domestic effluent is listed in. Revision is needed for "... rotten fruits and vegetables, that the major components of washing powder ..."

Table 2 and corresponding section: it is important to explain the expression "conception of pollutant via water". Is this referring to drinking water? It would be interesting to extend the literature review and provide, when possible, the admissible limits or the threshold concentrations of the listed pollutants leading to health hazards.

Table 2: The first column is labeled "Inorganic Elements”, but it lists molecules, organic compounds, bacteria, etc.

Table 2: the long list of pathogens (bacteria, viruses, etc.) is a copy of Table 2 in Chahal et al. (Adv Appl Microbiol. 2016; 97: 63–119). Do the authors have permission to do this? Similarly, what about Figure 6? The authors should make sure they have the permission to reproduce published data.

Line 192-194: the prediction of death tall due water-related diseases was for 2000 to 2020. The reader would expect the authors to follow up on this since we are in 2022.

Line 200: "... developing nations like India, ..., Africa ..." Africa is continent not a nation.

Lines 215-216: "It is believed that by 2025, India, China, and select countries of Europe and Africa will face water scarcity where Fig. 3 is showing world population." What is the role of Fig.3 here?

Figure 4 not called or discussed in the text. The 2nd and 3rd columns are both labeled “Population in Least Developed Countries”.

Line 223: a sentence should not start with a number (Eighty instead of 80).

Line 265: "Fig.6 is showing water cycle in urban region."

Lines 299-302: $25.3 billion do not account for 0.4% of a total investment of $230 billion.

Line 302: ... invested in water management?

Section 9: this should be the most important section, considering the title of this review (state of the art and emerging trends in the wastewater treatment). More literature review and development are needed here.

Author Response

In this manuscript, the authors compiled interesting information from the literature about wastewater. They described its sources, composition, impact on human health and treatment. The review was divided into 10 sections, only one of which (section 9) was focused, but not extensively, on wastewater treatment. In another section (8) the authors discussed the funding aspect. Based on the title, the manuscript should mainly and exhaustively review recent trends in wastewater treatment, which, unfortunately, was not the case. The text does not read well and contains many grammar errors. There is no smooth connection between sentences in many paragraphs (same as reading different notes from the literature). I would recommend an extensive revision of the text prior to publication.

  1. Abbreviations and acronyms used in the text were not defined. The authors should either define each abbreviation/acronym prior to first use in the text or provide a list of the abbreviations and their corresponding definitions.

A/R: Thank you for this valuable comment and suggestions. The authors have now defined the abbreviations and acronyms used in the text as suggested by the reviewer. The authors have also added a separate section at the end of the manuscript where all the abbreviations are introduced.

  1. Grammar revision is needed all along the text (e.g., verify text in lines 108-109, 110, 129-130, 152, 158, 161, 167-171, 198, 203, 232, 296, 311, 312, 326, 336, etc.)

A/R: Thank you for this valuable comment and suggestions. The authors have now grammatically rectified all such sentences as suggested by the reviewer in the revised manuscript.

  1. In some cases, “developing countries” was too much repeated in a same paragraph. For examples in lines 355-356, it reads in a row: “The developing countries should invest… The developing countries should actively participate …”. Some rewording and good grammar could help avoiding such unnecessary repetition.

A/R: Thank you for this valuable comment and suggestions. The authors have now rectified all such sentences in the revised manuscript as suggested by the reviewer.

  1. Since the review is about the "state of the art" of wastewater treatment, the section 2 is redundant.

A/R: Thank you for this valuable comment and suggestions. The authors have now deleted section 2 as per the suggestion of the reviewer in the revised manuscript as suggested by the reviewer.

  1. There are two figures 3 (Figure 3. Sources of Water Pollution, and Figure 3. Source – United Nation Population Fund.)

A/R: Thank you for this valuable comment and suggestions. The authors have now rectified the mistake in Fig. 3 as Fig 4 was by mistake written as Fig. 3.

  1. Lines 108-109: incomplete sentence

A/R: Thank you for this valuable comment and suggestions. The authors have now completed the said sentence in the revised manuscript as suggested by the reviewer.

  1. Line 110: "commercials" means advertisements. Also, instead of "... hospitals wastewater treatment and Industries.", I think you mean "... hospitals, wastewater treatment plants, and industries."

A/R: Thank you for this valuable comment and suggestions. The authors have now rectified the said sentences as per the suggestions of the reviewer in the revised manuscript.

  1. Line 111: "... run off from like run off from agriculture, ... ". Consider revision.

A/R: Thank you for this valuable comment and suggestions. The authors have now revised the sentence as per the suggestion of the reviewer.

  1. Table 1: title is about the composition of industrial effluent, but domestic effluent is listed in. Revision is needed for "... rotten fruits and vegetables, that the major components of washing powder ..."

A/R: Thank you for this valuable comment and suggestions. The authors have now rectified the title as well as the content of the table as suggested by the reviewer in the revised manuscript.

  1. Table 2 and corresponding section: it is important to explain the expression "conception of pollutant via water". Is this referring to drinking water? It would be interesting to extend the literature review and provide, when possible, the admissible limits or the threshold concentrations of the listed pollutants leading to health hazards.

A/R: Thank you for this valuable comment and suggestions. The authors have now rectified the mistake in the revised table as per the suggestions of the reviewer in the revised manuscript. The authors have now extended the literature review and provided the admissible limits or the threshold concentrations of the listed pollutants leading to health hazards in the revised manuscript.

  1. Table 2: The first column is labeled "Inorganic Elements”, but it lists molecules, organic compounds, bacteria, etc.

A/R: Thank you for this valuable comment and suggestions. The authors have now separated all such merged ones under a separate heading in the revised manuscript as suggested by the reviewer.

  1. Table 2: the long list of pathogens (bacteria, viruses, etc.) is a copy of Table 2 in Chahal et al. (Adv Appl Microbiol. 2016; 97: 63–119). Do the authors have permission to do this? Similarly, what about Figure 6? The authors should make sure they have the permission to reproduce published data.

A/R: Thank you for this valuable comment and suggestions. The authors have now modified the table and also took the copyright permission.

  1. Line 192-194: the prediction of death tall due water-related diseases was for 2000 to 2020. The reader would expect the authors to follow up on this since we are in 2022.

A/R: Thank you for this valuable comment and suggestions. The authors have now added the references up to 2022 in the revised manuscript as suggested by the reviewer.

  1. Line 200: "... developing nations like India, ..., Africa ..." Africa is continent not a nation.

A/R: Thank you for pointing out this mistake. The authors have now rectified the mistake in the revised manuscript as suggested by the reviewer.

  1. Lines 215-216: "It is believed that by 2025, India, China, and select countries of Europe and Africa will face water scarcity where Fig. 3 is showing world population." What is the role of Fig.3 here?

A/R: Thank you for pointing out this mistake. The authors have now rectified the mistake in the revised manuscript as suggested by the reviewer. The authors have changed Fig, 3.

  1. Figure 4 not called or discussed in the text. The 2ndand 3rd columns are both labeled “Population in Least Developed Countries”.

A/R: Thank you for pointing out this mistake. The authors have now rectified the mistake in the revised manuscript as suggested by the reviewer.

  1. Line 223: a sentence should not start with a number (Eighty instead of 80).

A/R: Thank you for pointing out this mistake. The authors have now rectified the mistake in the revised manuscript as suggested by the reviewer.

  1. Line 265: "Fig.6 is showing water cycle in urban region."

A/R: Thank you for pointing out this mistake. The authors have now rectified the mistake in the revised manuscript as suggested by the reviewer.

  1. Lines 299-302: $25.3 billion do not account for 0.4% of a total investment of $230 billion.

A/R: Thank you for pointing out this mistake. The authors have now rectified the mistake in the revised manuscript as suggested by the reviewer.

  1. Line 302: ... invested in water management?

A/R: Thank you for this comment and suggestion. The authors have now rectified the mistake in the revised manuscript as suggested by the reviewer.

  1. Section 9: this should be the most important section, considering the title of this review (state of the art and emerging trends in the wastewater treatment). More literature review and development are needed here.

A/R: Thank you for this valuable comment and suggestion. The authors have now emphasized has now on this section and elaborated in the revised manuscript as suggested by the reviewer.

Reviewer 3 Report

The Manuscript “The State of Art and Emerging Trends in The Wastewater Treatment in Developing Nationsneeds revision.

1.     This review failed to clearly identify major achievements in the field in recent years, major research questions, or future research needs, which are key aspects.

2.     Provide significant words which are more relevant to the work in a logical sequence as ‘keywords’, and also use keywords that are not present in the title.

3.     The "Introduction" section should follow the state of the art of this field and review what has been done, for supporting the research gap and the significance of this study. Please improve the state of the art overview, to clearly show the progress beyond the state of the art. The lack of proper justification creates the wrong impression that the authors are unaware of the recent developments. At the end of the introduction, the statement of the paper's goal and the explanation of novelty has to be properly formulated. Currently, this is not performed well. A high-quality paper has to provide a proper state-of-the-art analysis after the literature review and only based on the analysis to formulate the paper goals. The scientific basis and hypothesis for this study should be demonstrated in the "Introduction" section.

4.     The introduction of the review paper must be extended and reformulated in order to provide a more comprehensive approach.

5.     The last paragraph or closing lines of the introduction section (objectives) need detailed revision.

6.     It is also recommended to discuss and explain what should be the appropriate policies based on the findings of this study. Also, the literature should be further elaborated to show how they could be used for real applications.

7.     Have the authors rights to any figures they show in the manuscript?

Author Response

The Manuscript “The State of Art and Emerging Trends in The Wastewater Treatment in Developing Nations” needs revision.

  1. This review failed to clearly identify major achievements in the field in recent years, major research questions, or future research needs, which are key aspects.

A/R. Thank you for this valuable comment and suggestion. The authors have now talked about all the above-mentioned topics as suggested by the reviewer in the revised manuscript.

  1. Provide significant words which are more relevant to the work in a logical sequence as ‘keywords’, and also use keywords that are not present in the title.

A/R. Thank you for this valuable comment and suggestion. The authors have now modified the keywords as per the suggestion of the reviewer in the revised manuscript.

  1. The "Introduction" section should follow the state of the art of this field and review what has been done, for supporting the research gap and the significance of this study. Please improve the state of the art overview, to clearly show the progress beyond the state of the art. The lack of proper justification creates the wrong impression that the authors are unaware of the recent developments. At the end of the introduction, the statement of the paper's goal and the explanation of novelty has to be properly formulated. Currently, this is not performed well. A high-quality paper has to provide a proper state-of-the-art analysis after the literature review and only based on the analysis to formulate the paper goals. The scientific basis and hypothesis for this study should be demonstrated in the "Introduction" section.

A/R: Thank you for this valuable comment and suggestion. The authors have now thoroughly modified the introduction section as per the suggestion of the reviewer.

  1. The introduction of the review paper must be extended and reformulated in order to provide a more comprehensive approach.

A/R: Thank you for this valuable comment and suggestion. The authors have now thoroughly modified the introduction section as per the suggestion of the reviewer.

  1. The last paragraph or closing lines of the introduction section (objectives) need detailed revision.

A/R: Thank you for this valuable comment and suggestion. The authors have now added objectives in the revised manuscript as per the suggestion of the reviewer.

  1. It is also recommended to discuss and explain what should be the appropriate policies based on the findings of this study. Also, the literature should be further elaborated to show how they could be used for real applications.

A/R: Thank you for this valuable comment and suggestion. The authors have now discussed and explained the appropriate policies based on the findings of this study. Besides this author has also elaborated on how they could be used for real applications in the revised manuscript as suggested by the reviewer.

  1. Have the authors rights to any figures they show in the manuscript?

A/R: Thank you for this valuable comment and suggestion. Authors have now taken permissions for all the figures used in this manuscript.

Round 2

Reviewer 1 Report

All the comments and suggestions have been addressed carefully by authors, I have no other comments about this paper.

Reviewer 3 Report

Thank you for considering my comments.